# Translational rapid ultraviolet-excited sectioning tomography for whole-organ multicolor imaging with real-time molecular staining

**Wentao Yu, Lei Kang, Victor TC Tsang, Yan Zhang, Ivy HM Wong, Terence TW Wong\***

Translational and Advanced Bioimaging Laboratory, Department of Chemical and Biological Engineering, The Hong Kong University of Science and Technology, Hong Kong, China

**Abstract** Rapid multicolor three-dimensional (3D) imaging for centimeter-scale specimens with subcellular resolution remains a challenging but captivating scientific pursuit. Here, we present a fast, cost-effective, and robust multicolor whole-organ 3D imaging method assisted with ultraviolet (UV) surface excitation and vibratomy-assisted sectioning, termed translational rapid ultraviolet-excited sectioning tomography (TRUST). With an inexpensive UV light-emitting diode (UV-LED) and a color camera, TRUST achieves widefield exogenous molecular-specific fluorescence and endogenous content-rich autofluorescence imaging simultaneously while preserving low system complexity and system cost. Formalin-fixed specimens are stained layer by layer along with serial mechanical sectioning to achieve automated 3D imaging with high staining uniformity and time efficiency. 3D models of all vital organs in wild-type C57BL/6 mice with the 3D structure of their internal components (e.g., vessel network, glomeruli, and nerve tracts) can be reconstructed after imaging with TRUST to demonstrate its fast, robust, and high-content multicolor 3D imaging capability. Moreover, its potential for developmental biology has also been validated by imaging entire mouse embryos (~2 days for the embryo at the embryonic day of 15). TRUST offers a fast and cost-effective approach for high-resolution whole-organ multicolor 3D imaging while relieving researchers from the heavy sample preparation workload.

*For correspondence:
ttwwong@ust.hk

## Editor's evaluation

TRUST is a powerful content-rich three-dimensional imaging method. By combining iterative mechanical sectioning, automated labeling with fluorogenic dyes, UV-based surface illumination, and widefield multicolor detection, entire late gestation mouse embryos and terminally developed tissues can be imaged at cellular-scale resolution. Importantly, the strengths and weaknesses of the method are forthrightly presented. Ultimately, this work will be of great interest to developmental biologists, medical biologists, and clinicians, all of whom routinely work with specimens that are prohibitively large for labeling and imaging using classical mechanisms.

## Introduction

High-resolution three-dimensional (3D) whole-organ imaging with high fidelity has long been a scientific challenge. Mechanical sectioning followed by manual staining is a traditional approach for 3D histological imaging with almost no limitation on sample size *Amunts et al., 2013*. However, hundreds

to thousands of slices must be mounted on glass slides, stained with dyes, and then imaged by a conventional bright-field microscope with a tile scanning to cover the whole slide. The entire procedure is highly labor-intensive and requires complicated image registration algorithms *Wang et al., 2015*; *Ju et al., 2006* for 3D reconstruction. Moreover, sectioning before imaging could induce inevitable distortion in the reconstructed 3D image due to slice ruptures, which is impossible to be corrected by image processing approaches, hindering the broad applicability of this traditional method.

Over the past decade, many advanced tissue clearing protocols *Richardson and Lichtman, 2015*; *Richardson et al., 2021*; *Ueda et al., 2020* have been proposed to render opaque biological samples transparent and significantly reduce light scattering or absorption, thus facilitating volumetric imaging of thick tissue samples when combined with light microscopy approaches. Light-sheet fluorescence microscopy (LSFM) *Huisken et al., 2004* is ideal for imaging large cleared specimens with its high acquisition speed, high spatial resolution, and high light efficiency. For example, single-cell resolution or subcellular-resolution imaging over the entire mouse brain has been achieved within hours *Matsumoto et al., 2019* or days *Murakami et al., 2018*, respectively. Many issues concerning tissue clearing have been resolved or mitigated recently. For example, the toxicity of solvents, particularly benzyl alcohol and benzyl benzoate, can be addressed by using its alternatives, like ethyl cinnamate *Henning et al., 2019*, which also contributes to prolonged fluorescent protein emission *Klingberg et al., 2017*. Also, the active electrophoretic tissue clearing (ETC) device has been utilized in CLARITY *Chung et al., 2013* to accelerate the clearing process significantly. Perfusion-assisted agent release in situ *Yang et al., 2014* achieved much faster whole-body clearing without ETC, which may cause tissue degradation. In response to the potential tissue damage and molecular information loss, stabilization under harsh conditions via intramolecular epoxide linkages to prevent degradation *Park et al., 2018* was proposed using a flexible polyepoxide to preserve the tissue architecture in organ-scale transparent tissues.

Although both tissue clearing and LSFM are still under active development, barriers remain despite the significant accomplishment. For instance, the detective objective lens with a long working distance is generally required by LSFM when imaging large-volume samples, limiting the highest attainable resolution due to the consequent reduction in numerical aperture (NA). Besides, tissue clearing, especially for large samples, still need to balance the clearing effect and time cost to avoid tissue degradation. For example, aqueous techniques show relatively high biocompatibility, biosafety, and preservation of protein function, which, however, may take a longer time. Therefore, despite the destructive nature of the imaging process, block-face serial sectioning tomography has also gained much attention for automated and registration-free 3D imaging of large specimens. In this imaging and sectioning scheme, tissue clearing is not needed. More importantly, the imaging volume is relatively easy to be scaled up through engineering modification of the sectioning system, e.g., enlarging the sample holder. By far, spinning disk-based confocal microscopy *Seiriki et al., 2017*; *Seiriki et al., 2019*, two-photon microscopy *Economo et al., 2016*; *Ragan et al., 2012*, multiphoton microscopy *Abdeladim et al., 2019*, microscopy with ultraviolet surface excitation (MUSE) *Fereidouni et al., 2017*; *Guo et al., 2019*, optical coherence microscopy *Min et al., 2020*, and photoacoustic microscopy (PAM) *Wong et al., 2017* have been utilized for block-face serial sectioning tomography. However, there are other challenges in this field, including high system cost, time-consuming tissue processing (e.g., staining or embedding), and the slow imaging speed with point-scanning-based imaging systems. To this end, wide-field large-volume tomography (WVT) *Gong et al., 2016* proposed real-time labeling to reduce the time cost of sample staining significantly. However, the imaged sample must be embedded into a resin block after several days of processing to achieve fine sectioning by a microtome, which can also induce apparent tissue shrinkage due to dehydration *Rodgers et al., 2021*; *Gong et al., 2013* and is detrimental in high-resolution whole-organ imaging.

By far, numerous organ-scale imaging systems have been proposed with different focuses and strengths. With the goal of time-efficient, inexpensive, and multicolor large-volume 3D imaging, we developed translational rapid ultraviolet-excited sectioning tomography (TRUST) with the merits of both MUSE and WVT while bypassing the need for additional sample processing and keeping the entire system setup simple and cost-effective. More specifically, the short penetration depth of the UV light is utilized in TRUST for fast widefield block-face imaging. Meanwhile, formalin-fixed samples can be directly imaged by TRUST, and the tissue will be labeled layer by layer along with vibratomy-assisted serial sectioning. Compared with whole-organ pre-staining, real-time staining can

tremendously reduce the overall staining time, considering that the diffusion timescale correlates quadratically with the tissue thickness *Richardson et al., 2021*. Furthermore, the real-time staining is well fitted with the shallow penetration depth of the UV light.

While TRUST does achieve rapid 3D imaging with subcellular resolution, its unique strength lies in its high-content imaging ability with a cost-effective and straightforward setup. Oblique illumination using an inexpensive ultraviolet light-emitting diode (UV-LED) is implemented as the light source to simultaneously excite molecular-specific fluorescence and autofluorescence signals, which can then be collectively captured by a color camera. Many fluorescent dyes with different emission spectra can be excited simultaneously by the UV light *Fereidouni et al., 2017*, providing a broad and informative color palette. In TRUST, we mainly applied two fluorogenic dyes *Gao et al., 2020* (4',6-diamidino-2-phenylindole [DAPI] and propidium iodide [PI]) together for double labeling to provide high color contrast and reveal rich biological information (*Figure 1—figure supplement 1*). Nile red and DiI have also been applied for lipid staining in TRUST (*Figure 1—figure supplement 2*). To characterize the performance and examine the robustness of the proposed system, all vital organs in mice were imaged. To further show the full potential of TRUST, whole mouse embryos at different stages have also been imaged. The detailed image comparison demonstrates that TRUST is highly desirable for large sample imaging, showing great promise as a tool for developmental biology studies.

## Results

### TRUST system setup and workflow

In TRUST (*Figure 1a–c*), the short penetration depth in tissue of obliquely illuminated light from UV-LED (~285 nm) is the key to achieving widefield block-face imaging. Both fluorescence and autofluorescence signals from the tissue surface are excited by UV light, which are subsequently collected by a 10×infinity-corrected objective lens, and finally focused on a color camera by a tube lens. Two motorized stages (Motor-*x* and Motor-*y*) can drive the objective lens scanning along the *x-y* plane, and the manually tunable *z*-axis stage is used for focusing. A lab-built 2-axis angle adjustable platform (*Figure 1—figure supplement 3*) under the vibratome can keep the focal plane of the objective lens in parallel with the sample surface determined by the blade angle of the vibratome.

Fixed tissue block after agarose or gelatin embedding can be directly imaged with TRUST, and the sample will be stained during imaging by submerging it under staining solutions in the water tank of the vibratome. A 3D printed plastic waterproof case (*Figure 1c*) can reduce the fluorescence background by minimizing the volume of staining solutions filled between the tissue surface and objective lens (*Figure 1—figure supplement 4*). Besides, it can prevent the objective lens from being affected by liquid evaporation or fluctuation, ensuring high-quality imaging throughout the entire whole-organ imaging process. Two pieces of quartz mounted on the waterproof case are used to transmit the UV illumination beam and the excited fluorescence signal.

The workflow of the whole system (*Figure 1d*) can be simplified as a loop of three steps: (1) the surface layer of the sample immersed under chemical dyes will be labeled within ~150 s *Figure 1—figure supplement 5*; (2) the region of interest of the current layer, which consists of multiple fields of view (FOVs), will be acquired through motorized raster-scanning and subsequently stitched in parallel by a lab-built program; and (3) the vibratome will cut off the imaged layer and expose the layer underneath to the staining solutions. This loop will end when the whole organ has been imaged completely. To realize fully automated serial imaging, triggering circuits and corresponding control programs have been developed for synchronizing the entire TRUST system.

Because the sample is stained during the imaging step in TRUST, the labeling protocol can be thought of as real-time staining. Fluorogenic probes *Gao et al., 2020*; *Wang et al., 2020*; *Werther et al., 2021* which show an increase in fluorescence on binding to their targets, do not require a procedure to remove unbound probes and are naturally suited for the real-time staining owing to the low fluorescence background. Although lots of different fluorogenic probes have been synthesized for labeling various components, like proteins *Gao et al., 2020*; *Werther et al., 2021*; *Grimm et al., 2017*, lipid droplets *Fam et al., 2018*, cytoskeleton *Lukinavičius et al., 2014*, and mitochondrial *Fang et al., 2020*, two of the most commonly used and commercially available fluorogenic probes, DAPI and PI, were applied together in TRUST for nucleic acids staining. The fluorescence intensity of

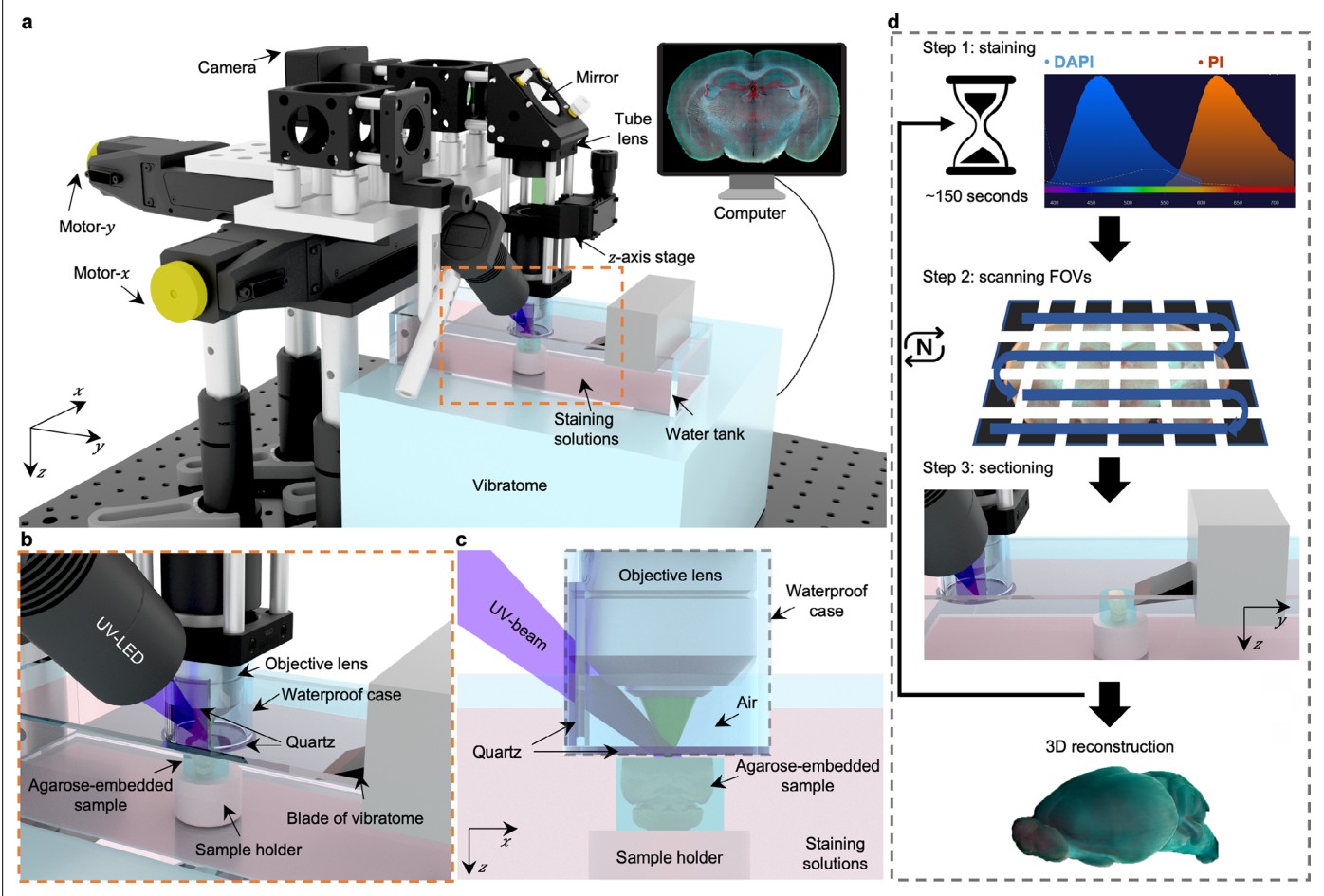

**Figure 1.** Overview of 3D whole-organ imaging by TRUST. (**a**) Schematic of the TRUST system. Light from UV-LED is obliquely projected onto the surface of an agarose-/gelatin-embedded sample (e.g., a mouse brain), which is placed on top of a sample holder inside a water tank of a vibratome filled with staining solutions. The generated fluorescence and autofluorescence signals are collected by an objective lens, refocused by an infinitely corrected tube lens, and finally detected by a color complementary metal-oxide-semiconductor camera. (**b**), Close-up of the region marked by orange dashed box in a, showing the components immersed in the staining solutions. (**c**), Viewing b from the x-z plane. A waterproof case containing two pieces of quartz keeps most of the space under the objective lens filled with air. (**d**), Workflow of the whole imaging process, including (1) chemical staining for ~150 s, (2) widefield imaging with raster-scanning and stitching in parallel, and (3) shaving off the imaged layer with the vibratome to expose a layer underneath. The three steps will be repeated until the entire organ has been imaged. All procedures are automated with lab-built hardware and control programs.

The online version of this article includes the following video and figure supplement(s) for figure 1:

**Figure supplement 1.** Comparison of single labeling (PI) and double labeling (DAPI and PI).

**Figure supplement 2.** TRUST images of brain and liver tissues stained with DiI and Nile red.

**Figure supplement 3.** The schematic of the customized 2-axis adjustable angular platform under the vibratome.

**Figure supplement 4.** High imaging contrast enabled by a waterproof case.

**Figure supplement 5.** Estimation of staining time by cell counting.

**Figure supplement 6.** Experimental characterization of TRUST's lateral resolution.

**Figure supplement 7.** Image quality comparison of TRUST images with deconvolution and surface extraction.

**Figure supplement 8.** Comparison of imaging results from 4×TRUST, 10×TRUST, and Deep-TRUST.

**Figure 1—video 1.** Recycling of sectioned slices with a pump.

https://elifesciences.org/articles/81015/figures#fig1video1

DAPI or PI will be amplified over 20 folds when bonded with the nucleic acids *Barcellona et al., 1990*; *Unal Cevik and Dalkara, 2003*, so that the fluorescence background from the staining solutions will be negligible. Finally, to maintain the concentration of chemical dyes in the water tank over a long period for whole-organ imaging, a high-precision water pump (LabN1-YZ1515x, Baoding Shenchen Precision Pump Co., Ltd) can be utilized to supply additional dyes into the water tank at a certain rate (*Supplementary file 1a*). Another peristaltic pump (KCP PRO2-N16, Kamoer Fluid Tech Co., Ltd.) with its suction pipe placed close to the sample holder of the vibratome can be utilized to collect the sectioned layers automatically for follow-up studies, like immunostaining, if necessary (*Figure 1— video 1*).

## Whole mouse brain imaging with TRUST

Our TRUST system first imaged a mouse brain to demonstrate its high imaging speed, molecular-specific real-time staining, and multicolor/multicontrast imaging capability. The mouse brain was first harvested and fixed in formalin for 24 hr. Subsequently, without staining or clearing, the mouse brain was directly embedded into 2% w/v agarose. The sectioning thickness of the vibratome was set as 50 μm to provide a good balance between the sectioning quality and the *z*-axis sampling interval. DAPI and PI solutions with a concentration of 5 μg/ml were used to label cell nuclei. The total imaging volume (12.1 mm × 8.6 mm×17.4 mm [*xyz*]) consists of approximately $7.8×10^{11}$ voxels with 24 bits RGB channels, of which the uncompressed dataset is ~2.1 terabytes (TB). Without image registration, images of 347 coronal sections acquired by TRUST can be directly stacked to reconstruct the 3D model (*Figure 2a*). Including staining, two-dimensional (2D) raster-scanning, and mechanical sectioning, the total acquisition time is ~64 hr, which is highly manageable and practical.

The real-time staining is well fitted with TRUST. Apart from the high time cost, traditional staining protocols *Seiriki et al., 2019* may also lead to uneven staining (*Figure 2—figure supplement 1*) due to the nature of passive diffusion. More importantly, agents like Triton X-100, which is used for increasing the permeability of tissue and accelerating the staining speed, can increase the transparency of the sample by washing away the lipids inside the tissue. Then, the imaging contrast of TRUST will be deteriorated due to the increased penetration depth of UV light (*Figure 2—figure supplement 1b–g*). In comparison, four coronal sections (*Figure 2c–f*) with positions indicated by the white dashed lines in *Figure 2b* show the stable sectioning performance of vibratome and uniform staining throughout the whole-brain imaging process.

When compared with other advanced 3D imaging systems, the advantages of TRUST are more related to its high-content multicolor imaging capability with an outstanding balance in terms of time-efficiency (sample preparation time and imaging speed), imaging resolution, and cost-effectiveness (*Supplementary file 1b*), by enjoying the benefits of the UV surface excitation and double labeling. Unlike the staining protocol used in MUSE or WVT, we developed a real-time double-labeling protocol that perfectly fits our TRUST system. The double labeling with DAPI and PI helps to reveal more biological information and achieve better image quality than that of staining with only PI (*Figure 1— figure supplement 1*). First, PI staining can clearly reveal cytomorphological details of neurons while the cell bodies of glial cells are only slightly labeled *Hezel et al., 2012*, hence differentiating neurons from glial cells in the brain through morphological differences (*Figure 1—figure supplement 1a*). In contrast, DAPI almost only stains cell nuclei. As a result, with the combination of DAPI and PI, TRUST can provide similar imaging contrast as Nissl stains (*Figure 2—figure supplement 2*), where the Purkinje cell layer can be easily differentiated from the granular layer or molecular layer. Furthermore, the color contrast of images can be improved by double labeling, especially for regions with a high density of cells (*Figure 1—figure supplement 1g and h*) because the cytoplasm stained by PI (red) can act as a background for DAPI-labeled cell nuclei to stand out (green and blue).

Although intrinsic autofluorescence is unwanted information in many cases, which can be regarded as background signals, much effort has been spent to reveal meaningful biological information in 2D label-free imaging *Ojaghi et al., 2020*; *Bhartia et al., 2010*; *Kaza et al., 2021*; *Yung et al., 2016*; *Costantini et al., 2021*, which can also be utilized in TRUST. For example, the axon (*Figure 2g*), nerve tracts (e.g., lateral olfactory tract *Figure 2h* or anterior commissure *Figure 2i*), and myelinated fiber bundles in the caudate putamen (*Figure 2j*) have been identified with TRUST even without any labeling. Also, blood vessels (*Figure 2k*) can be identified based on negative contrast because their autofluorescence signal intensity is lower than that of the surrounding tissues *Mehrvar et al.,*

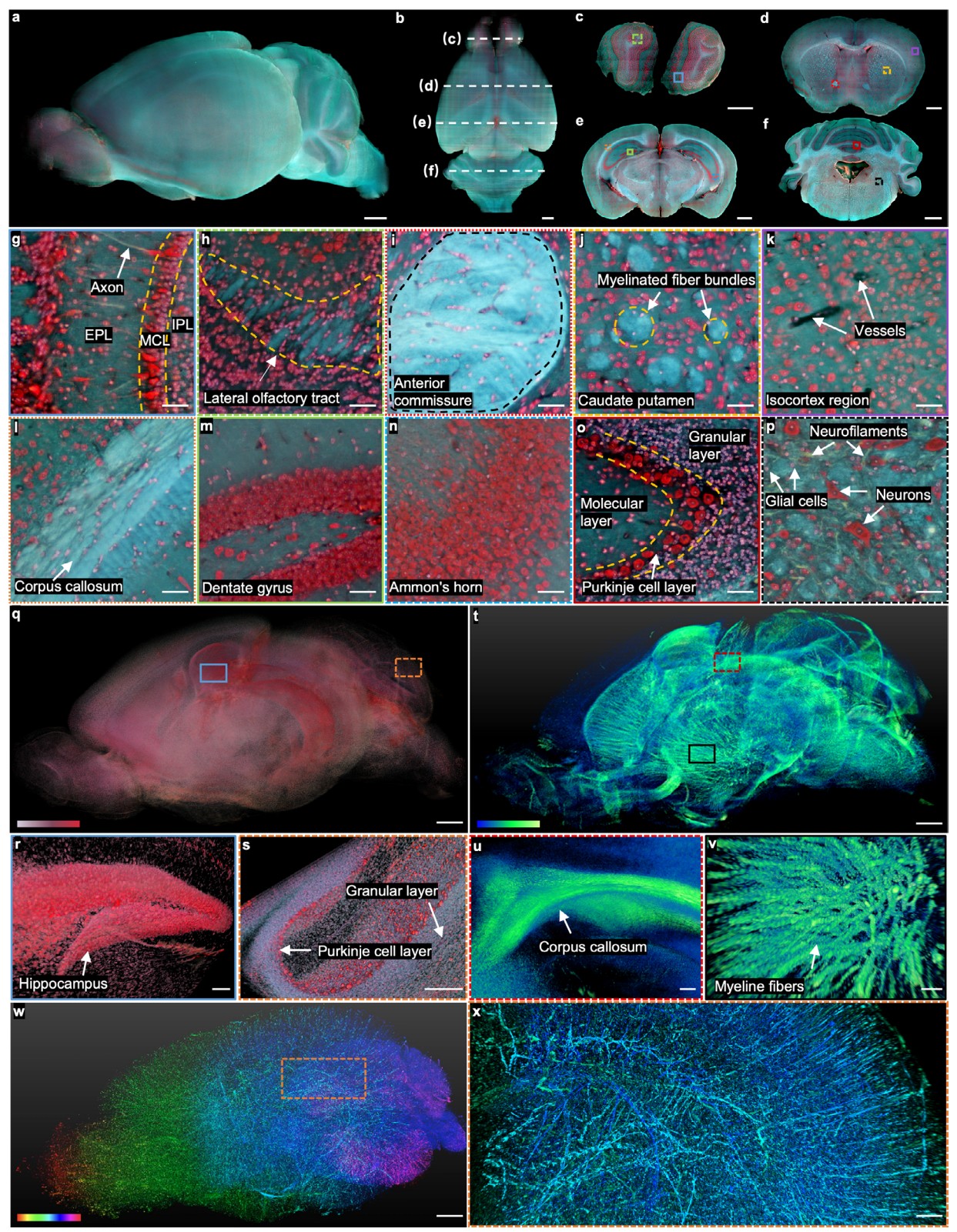

**Figure 2.** Whole mouse brain imaging with TRUST. (**a, b**), Side and top views of the reconstructed 3D model of a fixed mouse brain, respectively. (**c–f**) Four coronal sections with positions indicated in b by the white dashed lines. (**g**), (**h**), Close-up images of the blue solid and green dashed regions marked in c. (**i–k**), Close-up images of the red dotted, yellow dashed, and purple solid regions marked in d. (**l–n**), Close-up images of the orange dotted, green solid, and blue dashed regions marked in e.( **o**), (**p**), Close-up images of the red solid and black dashed regions marked in f. (**q**), The cell nuclei in

*Figure 2 continued on next page*

*Figure 2 continued*

the mouse brain extracted from the red channel in a. (**r**), ( **s**), Close-up images of the blue solid and orange dashed regions marked in q. (**t**), 3D structure of the nerve tracts or fibers extracted from the green channel of the autofluorescence signal in a. (**u**), (**v**), Close-up images of the red dashed and black solid regions marked in t. (**w**), Vessel network extracted from another mouse brain and rendered with different colors corresponding to different coronal layers. (**x**), Close-up image of the orange dashed region marked in w. MCL: mitral cell layer; IPL: inner plexiform layer; EPL: external plexiform layer. Scale bars: 1 mm (**a–f, q,t,w**), 200 μm (**x**), 100 μm (**r–v**), and 50 μm (**g–p**).

The online version of this article includes the following video and figure supplement(s) for figure 2:

**Figure supplement 1.** Potential side effects of conventional chemical staining protocol *Seiriki et al., 2019*.

**Figure supplement 2.** Double labeling (DAPI and PI) can serve as Nissl staining.

**Figure supplement 3.** Cerebral vascular imaging with the assistance of intravascular perfusion.

**Figure supplement 4.** Double labeling (DAPI and PI) provides high molecular contrast from the autofluorescence background.

**Figure supplement 5.** TRUST images of a fresh mouse brain.

**Figure 2—video 1.** 3D rendering of a whole mouse brain.

https://elifesciences.org/articles/81015/figures#fig2video1

**Figure 2—video 2.** 3D rendering of the vessel network in a mouse brain.

https://elifesciences.org/articles/81015/figures#fig2video2

---

*2021*; *Staniszewski et al., 2013*. To further enhance the contrast of the vessel network in the brain (*Figure 2w and x*), the mouse was perfused transcardially with a mixture of black ink and 3% w/v gelatin (*Figure 2—figure supplement 3*).

In TRUST, the mixed dyes with a broad emission spectrum (blue to red) make the fluorescence signals less affected by the autofluorescence background by extracting signals from different color channels for different types of tissue components. For example, although hepatocytes in liver tissue are hard to be differentiated from the autofluorescence background in the red channel, the image contrast is high in the green and blue channels (*Figure 2—figure supplement 4a–d*). Also, although the autofluorescence background is strong in the green and blue channels for some regions in the mouse brain, the cells remain evident in the red channel (*Figure 2—figure supplement 4e–h*). With background subtraction and dynamic range adjustment in different color channels of TRUST images, the 3D distribution of cell nuclei (*Figure 2—figure supplement 1q-s*) and nerve fiber bundles (*Figure 2t–v*) in the brain can be digitally extracted. The 3D animation of the whole mouse brain, including serial coronal or sagittal sections, has been rendered as shown in *Figure 2—video 1*. The vessel network has also been rendered, as shown in *Figure 2—video 2*.

## 3D imaging of other organs with TRUST

To demonstrate the generalizability and robustness of the TRUST system, other mouse organs with various sizes are imaged, including a heart (*Figure 3a–e*), liver (*Figure 3f–j*), kidney (*Figure 3k–o*), lung (*Figure 3p–t*), and spleen (*Figure 3u–y*). A fixed mouse heart was first imaged by TRUST with a sectioning thickness of 50 μm. The whole imaging and staining automated procedure for the entire volume (10.4 mm × 8.2 mm × 6.1 mm, $1.8 \times 10^9$ voxels with 24 bits RGB channels, 122 sections) took ~21 hr, and the reconstructed 3D model is shown in *Figure 3a*. One representative section is shown in *Figure 3b* with its position marked by a white dashed line at the top right-hand corner. Three close-up images (*Figure 3c–e*) indicate that not only the cell nucleus, but other components, like adipose tissue (*Figure 3c*) or cardiac muscle (*Figure 3d*), can also be clearly imaged. More details of the whole dataset, including multiple zoomed-in regions, can be found in the rendered 3D animation (*Figure 3—video 1*).

Part of a fixed mouse liver (*Figure 3f*) has also been imaged by TRUST with a sectioning thickness of 50 μm. The entire imaging and staining process for the total volume (8.8 mm × 12.5 mm × 3 mm, $1.4 \times 10^{11}$ voxels with 24 bits RGB channels, 60 sections) took ~11 hr. The curved white dashed line marked in *Figure 3f* indicates the position of the section in *Figure 3g*. Two close-up images (*Figure 3h and i*) of the white solid and yellow dashed regions in *Figure 3g* show that hepatocytes and typical anatomical structures, like portal vein, sinusoids, and focal inflammation, can be well differentiated with TRUST. Based on the negative contrast of blood vessels, as shown in *Figure 3i*, the vessel network (*Figure 3j*) of the whole dataset can also be extracted with several image processing steps (*Figure 3—figure supplement 1*). Besides, by extracting features through different color channels

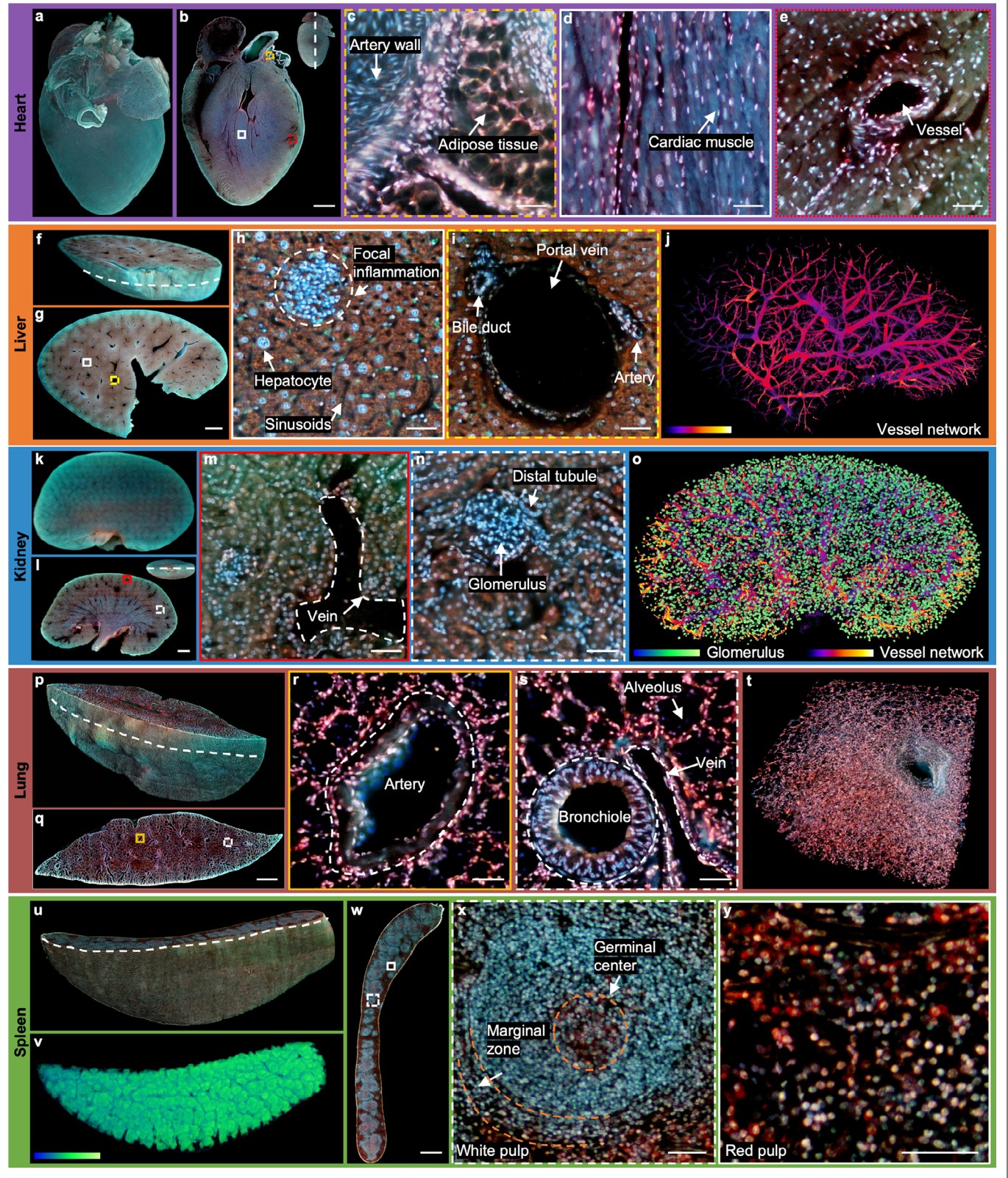

**Figure 3.** 2D/3D image gallery of other organs in mice with TRUST. (**a**), Reconstructed 3D model of the whole mouse heart. (**b**), One section of the heart with the position indicated by the white dashed line at the top right-hand corner. (**c–e**), Three close-up images of the orange dashed, white solid, and red dotted regions marked in b. (**f**), One block of a fixed mouse liver. (**g**), One section of the liver with the position indicated by the curved white dashed line marked in f. (**h**), (**i**), Close-up images of the white solid and yellow dashed regions marked in g. (**j**), Vessel network extracted based on negative

*Figure 3 continued on next page*

*Figure 3 continued*

contrast and rendered with pseudo color. (**k**), Reconstructed 3D model of a whole mouse kidney. (**l**), One section in the middle of the kidney with the position indicated by the white dashed line at the top right-hand corner. (**m**), (**n**), Close-up images of the red solid and white dashed regions marked in l. (**o**), Vessel network and glomeruli extracted from the whole kidney and rendered with pseudo color. (**p**), Part of a mouse lung imaged with TRUST. The curved white dashed line indicates the position of the section in q.( **r**), (**s**), Close-up images of the orange solid and white dashed regions marked in q. (**t**), A small zoomed-in region in p.( **u**), Rendered 3D model of a mouse spleen, and the curved white dashed line indicates the position of the section in w. (**v**), White pulp regions extracted from u through the blue channel and rendered with pseudo color. (**x**), (**y**), Close-up images of the white dashed and white solid regions marked in w. Scale bars: 1 mm (**b,g,l,q,w**) and 50 µm (**c–e, h**), (**i**), (**m**), (**n**), (**r**), (**s**), (**x**), (**y**).

The online version of this article includes the following video and figure supplement(s) for figure 3:

**Figure supplement 1.** Extraction of the vascular network.

**Figure supplement 2.** Nucleus and cytoplasm extraction from a TRUST image.

**Figure supplement 3.** A mouse liver block imaged by TRUST with 10 µm *z*-sampling intervals.

**Figure supplement 4.** Estimation of the penetration depth of UV light by cell counting.

**Figure supplement 5.** Gelatine perfusion of a mouse lung for high-quality sectioning.

**Figure 3—video 1.** 3D rendering of a whole mouse heart.
https://elifesciences.org/articles/81015/figures#fig3video1

**Figure 3—video 2.** 3D rendering of a whole mouse liver.
https://elifesciences.org/articles/81015/figures#fig3video2

**Figure 3—video 3.** 3D rendering of a whole mouse kidney.
https://elifesciences.org/articles/81015/figures#fig3video3

**Figure 3—video 4.** 3D rendering of a whole mouse lung.
https://elifesciences.org/articles/81015/figures#fig3video4

**Figure 3—video 5.** 3D rendering of a whole mouse spleen.
https://elifesciences.org/articles/81015/figures#fig3video5

and subsequent image binarization, cytoplasm and the cell nucleus can be separated to estimate the nuclear/cytoplasmic ratio (*Figure 3—figure supplement 2*). 3D animation of the whole dataset has been rendered as shown in *Figure 3—video 2*. Moreover, another block of the liver with a volume of 500 µm × 500 µm × 250 µm (*Figure 3—figure supplement 3*) has also been imaged by TRUST with a sectioning thickness of 10 µm, which is also matched with the penetration depth of UV light (*Figure 3—figure supplement 4*). A finer sectioning thickness results in a decrease in the *z*-sampling interval, which is helpful for resolving microstructures, such as the sinusoids in 3D space.

Imaging for the whole mouse kidney (*Figure 3k*) took~19hr and the sectioning thickness of the volume (13.2 mm × 8.6 mm×4.7 mm, $2.3 \times 10^{11}$ voxels with 24 bits RGB channels, 93 sections) was also set as 50µm. One typical section is shown in *Figure 3l* with its position marked by the white dashed line at the top right-hand corner. Vein and glomerulus can be clearly recognized as shown in *Figure 3m and n*, respectively. All glomeruli in the kidney were extracted with the assistance of an open-source state-of-the-art detection and segmentation machine learning library, Detectron2 *Wu et al., 2019*. The reconstructed vessel network together with the glomeruli of the whole kidney is shown in *Figure 3—video 1* of the whole dataset has been rendered as shown in *Figure 3—video 3*.

Sectioning lung tissue directly with a vibratome can be problematic due to porous structures, e.g., alveoli or bronchioles (*Figure 3—figure supplement 5a and b*). To achieve better sectioning performance, we first filled the lung with 10%w/v melted gelatin by syringe injection through its trachea, and the processed lung sample should be cooled down quickly and moved into formalin solution at 4° C overnight for post-fixation (*Figure 3—figure supplement 5c*). The entire imaging and staining process for the total volume (15.4 mm × 4.7 mm × 4.6 mm, $1.4 \times 10^{11}$ voxels with 24 bits RGB channels, 91 sections) took~11hr with a sectioning thickness of 50µm, and its reconstructed 3D model is shown in *Figure 3p*. To exhibit the improved sectioning performance, all sections of the whole dataset including multiple zoomed-in regions have been rendered in the 3D animation video (*Figure 3—video 4*). Two close-up images (*Figure 3r and s*) of the yellow solid and white dashed regions marked in *Figure 3q* show that the common anatomical structures, including artery, bronchiole, and alveolus, can be well imaged.

The final whole organ that we imaged was a mouse spleen. The entire imaging and staining process for the total volume (12.7 mm × 8.2 mm×3.8 mm, 1.7×10$^{11}$ voxels with 24 bits RGB channels, 75 sections) of the mouse spleen embedded in gelatin block took ~14 hr. The reconstructed 3D model is shown in *Figure 3u*, and the curved white dashed line indicates the position of the section shown in *Figure 3w*. Two close-up images (*Figure 3x and y*) of the white dashed and white solid regions marked in *Figure 3w* correspond to two major components in the spleen: white pulp and red pulp, respectively. Typical structures, like germinal center or marginal zone, can be clearly differentiated. As the overall appearance of the white pulp regions (*Figure 3x*) look blue, as shown in *Figure 3v*, they can be extracted from the blue channel with simple thresholding. Finally, 3D animation of the whole dataset also has been rendered, as shown in *Figure 3—video 5*.

## 2.4. 3D imaging of whole mouse embryos with TRUST

Under embryonic development, there are dramatic changes in cell/structural morphology and arrangement. Therefore, 3D imaging of a whole mouse embryo is vitally essential to assist biologists in understanding any anatomical and functional changes. However, imaging the whole embryo is still a scientific challenge and suffers from issues like low imaging resolution (micro-CT *Wong et al., 2012* and optical projection tomography *Ban et al., 2019*) or a limited number of sections (histological imaging *Crawford et al., 2010*; *Chen et al., 2017*). In the above, we have already demonstrated the advantages of our TRUST system, especially its ability for handling various types of organs regardless of their sizes, and its high imaging speed due to the real-time staining and widefield imaging configuration. Therefore, the TRUST system theoretically has the potential to address the needs of developmental biology.

We first imaged two heads from two mouse embryos with intact skulls (embryonic day (E) 15: *Figure 4a–f*; E18: *Figure 4g–l*). The whole imaging procedure (including staining, scanning, and sectioning) for the embryo head at E15 (9.9 mm × 9 mm × 6.9 mm, 2.6×10$^{11}$ voxels with 24 bits RGB channels, 137 sections) took ~20 hr, and the reconstructed 3D model from different viewing angles is shown in *Figure 4a*. The skin outside the embryo head at E18 was manually removed to achieve better sectioning performance with the vibratome. It took ~32 hr for both imaging and staining (9.4 mm × 7.8 mm × 10 mm, 3.1×10$^{11}$ voxels with 24 bits RGB channels, 201 sections). Comparing the two datasets makes it possible to determine the evolution of main morphological features over different developing stages. For example, close-up images of the retina areas from the two embryos are compared, as shown in *Figure 4e* (E15) and *Figure 4k* (E18). It can be observed that the inner plexiform layer (IPL) becomes apparent only in the E18 embryo, which is consistent with the neural differentiation status of the retinal region *Fan et al., 2016*. Also, *Figure 4f* and *Figure 4l* correspond to the cerebral cortex regions in the brain of the E15 embryo and E18 embryo, respectively. The ventricular zone (VZ) is easier to be differentiated from the intermediate zone (IZ) in the E18 embryo due to the differentiation of cortical neuroectoderm *Savolainen et al., 2009*. Similar results have also been observed in histological images *Chen et al., 2017*, proving the images from TRUST are highly reliable.

Then, we performed 3D imaging of two whole mouse embryos using our TRUST system with a sectioning thickness of 50 μm. The total imaging and staining process for the E15 embryo (9.9 mm × 7.8 mm × 12.8 mm, 4.2×10$^{11}$ voxels with 24 bits RGB channels, 256 sections) took ~2 days (*Figure 4m–r*). The skin of the E18 embryo was also removed, and the entire imaging and staining time (13.2 mm × 9.8 mm × 16.8 mm, 9.3×10$^{11}$ voxels with 24 bits RGB channels, 336 sections) took ~3 days (*Figure 4—figure supplement 1s-x*). The acquired datasets clearly visualized the morphology of whole mouse embryos and can be further processed to deliver quantitative 3D information. For example, a quantitative volume comparison between embryos at different stages has been made in *Figure 4—figure supplement 1*. Also, anatomical differences can be found between the E15 and E18 embryos. For example, close-up images of the cartilage primordium regions from the E15 embryo (*Figure 4p*) and E18 embryo (*Figure 4v*) show noticeable anatomical differences due to the ossification. Also, the morphological difference between the submandibular gland in the E15 embryo (*Figure 4q*) and E18 embryo (*Figure 4w*) is consistent with the fact that the acini in a mouse embryo will not complete lumenization until E17 *Larsen et al., 2010*. Finally, because of differentiation, cell nuclei in the thymic rudiment regions of the E15 embryo (*Figure 4r*) are obviously larger than that in the E18 embryo (*Figure 4x*), which is also consistent with the histological images in eHistology Atlas *Graham et al., 2015* (*Figure 4—figure supplement 2*).

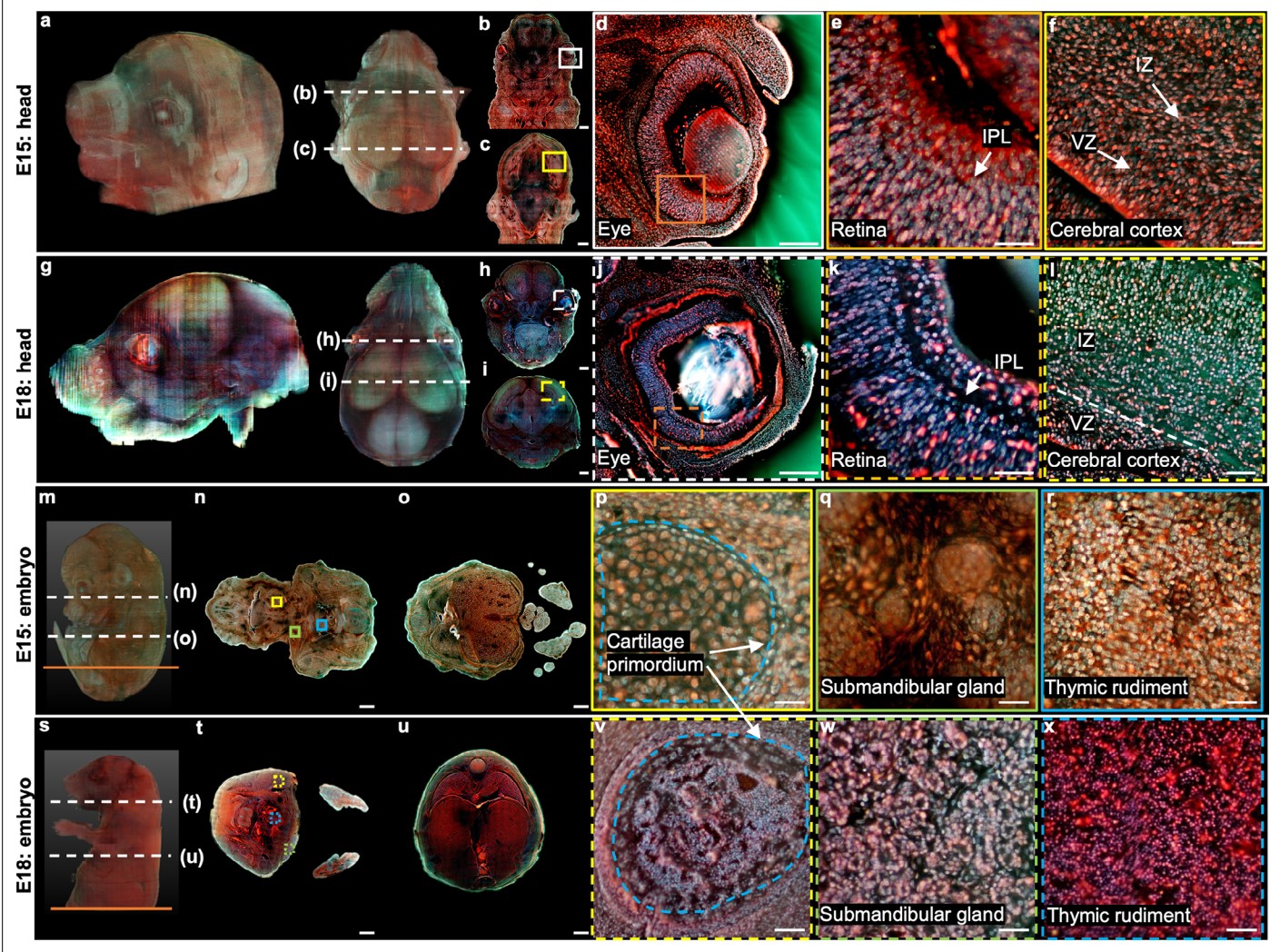

**Figure 4.** 3D imaging of mouse embryos. (**a**), Rendered 3D model of the embryo head at E15 viewed from two perspectives. The two white dashed lines indicate the positions of two sections in b,c. (**d**) Close-up image of the white solid region marked in b. (**e**) Close-up image of the orange solid region marked in d. (**f**) Close-up image of the yellow solid region marked in c. (**g**) Rendered 3D model of the embryo head at E18 viewed from two perspectives. The two white dashed lines indicate the positions of two sections in h,i. (**j**) Close-up image of the white dashed region marked in h. (**k**) Close-up image of the orange dashed region marked in j. (**l**) Close-up image of the yellow dashed region marked in i. (**m**) Rendered 3D model of a whole mouse embryo at E15. The two white dashed lines indicate the positions of two sections in n,o. (**p–r**), Close-up images of the yellow, green, and blue solid regions marked in n. (**s**) Rendered 3D model of a whole mouse embryo at E18. The two white dashed lines indicate the positions of two sections in t,u. (**v–x**) Close-up images of the yellow, green, and blue dashed regions marked in t. Scale bars: 1 mm (**b,c,h,i,n,o,t,u**), 500 μm (**d,j**), and 50 μm (**e,f,k,l,p–r,v–x**).

The online version of this article includes the following figure supplement(s) for figure 4:

**Figure supplement 1.** Volume quantification and comparison between embryos at different developmental stages.

**Figure supplement 2.** Comparison of TRUST and histological images of the thymic rudiment.

To better illustrate the difference between the two embryos, two whole sections from two embryos with positions marked by the orange solid lines in *Figure 4m* and *Figure 4—figure supplement 1s* are shown in *Figure 5*. More specifically, *Figure 5a1* and *Figure 5a2* are the zoomed-in regions of the dorsal root ganglion near the spinal cord. The cell nuclei in the latter look relatively small due to cell differentiation. *Figure 5b1* and *Figure 5b2* are the close-up images of the kidney area. Many ureteric buds and almost no glomeruli are observed in the E15 embryo, while glomeruli are easily observed in the E18 embryo. The result is consistent with the fact that the formation of glomeruli approximately begins at E13.5, with its number subsequently increasing up to 54 folds from E13.5 to E17.5 *Yi et al., 2010*. *Figure 5c1* and *Figure 5c2* are the zoomed-in regions of the spinal cord. The central canal in

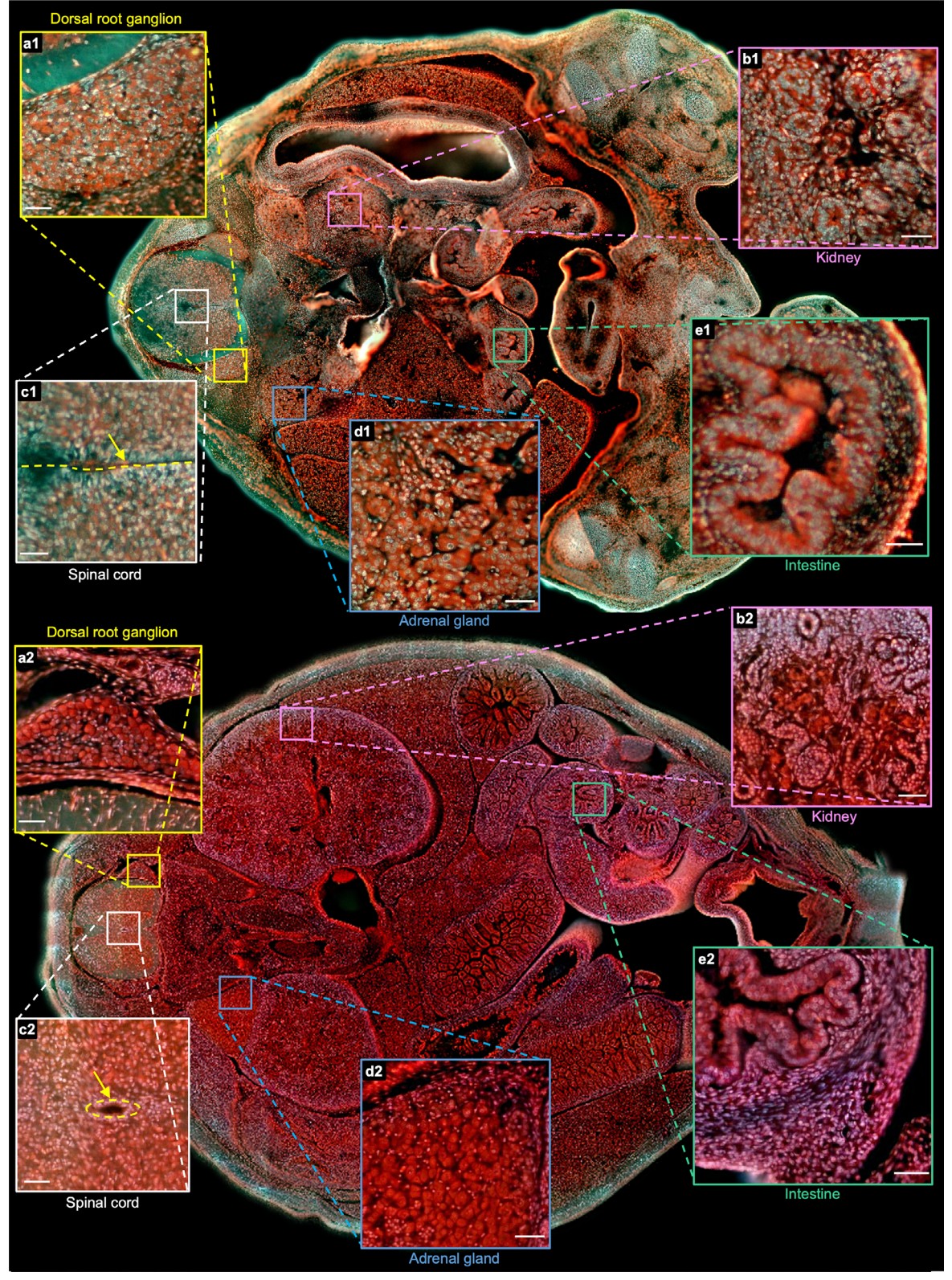

**Figure 5.** Comparison of two sections from the E15 and E18 embryos. a1(a2)–e1(e2), Close-up images from the E15(E18) embryo of the dorsal root ganglion, kidney, spinal cord, adrenal gland, and intestine marked with yellow, pink, white, blue, and green solid boxes, respectively. Scale bars: 50 μm.

the former is larger than that of the latter, which has also been observed through histological imaging *Chen et al., 2017*. *Figure 5d1* and *Figure 5d2* are the zoomed-in regions of the adrenal gland. Due to cell differentiation, the cell nuclei in the former are obviously larger than that of the latter. *Figure 5e1* and *Figure 5e2* are the zoomed-in regions of the intestines. The muscle structure in the latter one is more evident than that of the former one. These high-resolution and rich-molecular contrast TRUST images show that TRUST is an ideal tool for basic biology studies.

## Discussion

In summary, TRUST provides high-content multicolor 3D imaging with subcellular resolution in a time- and cost-effective manner by eliminating the need for sample preparation steps (e.g., dehydration, delipidation, decalcification, or pre-staining) and utilizing simple oblique widefield illumination with an inexpensive UV-LED for block-face imaging. UV light excited exogenous molecular-specific fluorescence and endogenous content-rich autofluorescence signals can be collectively captured by a color camera, providing a broad and informative color palette. The entire brain and all vital organs from adult mice have been imaged by TRUST with high quality and free of distortion, demonstrating its high performance and robustness. Also, the structural and morphological comparison of the two sets of images from two whole mouse embryos at the developing stages of E15 and E18 shows that TRUST is a promising tool for embryonic developmental study.

TRUST is at its early developing stage. To further improve the performance of the system, future research development can be carried out in the following aspects: First, the imaging resolution of TRUST is anisotropic. The lateral resolution of TRUST with a 0.25 NA objective lens is ~1.25 μm (*Figure 1—figure supplement 6*). However, the axial resolution of TRUST can be 10–50 μm, which is a major limitation when compared with other 3D imaging techniques *Matsumoto et al., 2019*; *Ragan et al., 2012*; *Wong et al., 2017*; *Gong et al., 2016*; *Gong et al., 2013*; *Li et al., 2010*; *Supplementary file 1b*, caused by (1) the sectioning thickness of the vibratome or (2) the optical-sectioning thickness provided by UV surface excitation. For the sectioning thickness, although a small block of the liver has been successfully imaged by TRUST with a 10 μm interval, to produce even and consistent sections, the sectioning thickness for other samples is all set as 50 μm. In the future, a more advanced vibratome with better performance (e.g., VF-300-0Z, Precisionary Instruments Inc) can be used to alleviate this problem. The optical-sectioning thickness provided by the UV surface excitation (~10–20 μm *Fereidouni et al., 2017*; *Guo et al., 2019*; *Wong et al., 2017*; *Yoshitake et al., 2018Guo et al., 2019*) can be decreased to ~1 μm *Guo et al., 2019* by doping the sample with UV-absorbing dyes. However, this method will impose more stringent requirements on mechanical sectioning. Therefore, as shown in *Figure 1—figure supplement 7*, using an objective lens with high NA and applying a deconvolution algorithm (e.g., Lucy–Richardson deconvolution *Lucy, 1974*) could be a simple and sufficient way to improve the lateral and axial resolutions while relaxing the requirement of the mechanical sectioning.

Besides, the total scanning time ($T_0$) for each section of the TRUST system can be further reduced, which is calculated as:

$$T_0 = (t_e + t_s) \times N$$

where $t_e$ is the exposure time for each FOV, $t_s$ is the motor scanning time required to move from the first FOV to the next FOV, and $N$ is the total number of FOVs. Currently, $t_e$ is relatively large (~250ms) to maintain an acceptable SNR. Employing multiple light sources (e.g., four UV-LEDs) and illuminating the samples from different angles simultaneously *Xie et al., 2019* is a feasible solution to decrease the $t_e$ by a factor of two. Also, $t_s$ can be reduced, at the least, by a factor of two through updating the current motor (20 mm/s; L-509, PI miCos GmbH) to an advanced model (90 mm/s; L-511, PI miCos GmbH). Moreover, the FOV is relatively small (600 μm × 450 μm) with the current camera (DS-Fi3, Nikon Corp.; chip size: 6.91 mm × 4.92 mm) and 10×objective lens. Upgrading the camera with a large chip size (e.g., GS3-U3-123S6C-C, Edmund Optics Inc; chip size: 14.13 mm × 10.35 mm) and using a low magnification (e.g., 4×) objective lens can reduce the $N$ by a factor of ~25. As for the degraded imaging resolution, it can be later recovered with the help of deep-learning-based super-resolution networks (*Wang et al., 2021*; *Figure 1—figure supplement 8*). In summary, $T_0$ can

be potentially reduced by a factor of ~50 with comparable image quality while preserving the high cost-effectiveness of the system.

Also, fresh tissue 3D imaging with serial sectioning and real-time staining is one promising working direction that can maximize one advantage of TRUST: free of sample processing. Although preliminary imaging results of the fresh mouse brain with real-time staining by Hoechst 33342 at the concentration of 20 µg/ml have been acquired (*Figure 2—figure supplement 5*), a more advanced vibratome (e.g., VF-300-0Z, Precisionary Instruments Inc) specifically for fresh tissue sectioning is needed in the future to improve the sectioning quality. Deconvolution and extended depth of focus algorithms applied in *Figure 1—figure supplement 7* may also be applied together to relieve the issue.

In addition, by leveraging the fact that TRUST provides exogenous molecular-specific fluorescence and endogenous content-rich autofluorescence imaging simultaneously, the high-content imaging capability of TRUST can be further improved. Firstly, more types of fluorogenic probes targeting components other than the cell nucleus can further enrich the imaging contents of TRUST. For example, protein (8-anilino-1-naphthalene sulfonic acid, Sigma-Aldrich) and lipids (Nile red or DiI, Sigma-Aldrich, *Figure 1—figure supplement 2*) can also be labeled. Besides, immunostaining, as one of the most powerful techniques for labeling cells/tissue with high molecular specificity, could be directly applied to each sectioned layer for a follow-up study because vibratome sectioning does not involve any harsh organic solvents. Also, minor modifications can be made to the current TRUST system in order to automate the whole immunostaining process. For example, prior to imaging the exposed surface layer of the tissue block, several water pumps can be used to vary the chemical solutions in the water tank of the vibratome to mimic the general staining procedures applied in immunostaining. To reduce the number of antibodies needed, a spray can be utilized to deliver the antibody onto the surface of the sample, instead of directly submerging the sample into the antibody solutions. Second, autofluorescence spectra of some tissue components can overlap with each other *Jamme et al., 2013*, making them hard to be differentiated. Automatically switching between different spectral windows according to the desired imaging targets by electrically tunable band-pass filters (e.g., KURIOS-VB1, Thorlabs Inc) is one possible solution, despite the exposure time for each FOV could increase because of time multiplexing.

Finally, the staining protocols can be optimized. Choosing dye with high fluorescence enhancement factors upon binding to its target, e.g., SYTOX Blue (Thermo Fisher Scientific Inc, 500-fold improvement) and YOYO-1 (Thermo Fisher Scientific Inc,>1000 fold improvement), can further decrease the background signal caused by the staining solutions. Another benefit that comes with it is that the concentration of the staining solutions can be further increased to accelerate the staining speed. Also, the speed control of adding additional dyes with the pump for maintaining the concentration of the stains can be more accurate through negative feedback: the pumping speed will be automatically adjusted with the change of average brightness of each 2D TRUST image.

## Methods
### TRUST imaging system

The short penetration depth of UV light is utilized for 2D block-face imaging. By combining with a vibratome, the TRUST system can realize serial surface imaging followed by sectioning for the whole sample. The schematic of the TRUST system is shown in *Figure 1a–c*. Firstly, UV light from a mounted LED (M285L5, Thorlabs Inc) is slightly focused on the surface of the specimen by a pair of lenses (#67–270, Edmund Optics Inc; LA4306-UV, Thorlabs Inc) with an oblique orientation. Then, excited fluorescence and autofluorescence signals will be collected by an objective lens (Plan N 10× / 0.25 NA, Air, RMS10X, Olympus Corp.), refocused by an infinity-corrected tube lens (ACA254-200-A, Thorlabs Inc), reflected by an aluminum mirror (PF10-03-G01, Thorlabs Inc), and finally detected by a color camera (DS-Fi3, Nikon Inc). An optical long-pass filter can be placed before the tube lens to further reduce the backscattered UV light. A custom-built 2-axis angle adjustable platform (*Figure 1—figure supplement 3*) under the vibratome (VF-700-0Z, Precisionary Instruments Inc) ensures the focal plane of the objective lens is parallel with the surface of the sample sectioned by the vibratome. The selected vibratome can slice samples up to 20 mm in diameter, and another model (VF-800-0Z, Precision Instruments Inc) can be used when imaging and slicing even larger samples (up to 100 mm in diameter). Two motorized stages (Motor-x: L-509.20SD00, Motor-y: L-509.40SD00; PI miCos GmbH) and a manually

tunable *z*-axis stage are used for 2D raster-scanning and objective lens focusing, respectively. A 3D printed plastic waterproof case (*Figure 1c*) can reduce the fluorescence background from the staining solutions, hence improving the imaging contrast (*Figure 1—figure supplement 4*), by keeping most of the space between the objective lens and sample surface filled with air instead of staining solutions. A waterproof case can also keep the imaging plane uninfluenced by the fluid level fluctuations induced by mechanical scanning and liquid evaporation. Two pieces of quartz mounted on the waterproof case are used for the transmission of both UV light and excited signals.

The workflow of the whole imaging system is basically the repetition of three steps as shown in *Figure 1d*: Step (1) the surface layer of the sample with an optical-sectioning thickness provided by UV surface excitation will be quickly stained by the chemical dyes (DAPI and PI) with ~150 s; Step (2) during imaging mode, the motor-*y* will move rightwards until the objective lens is located above the imaged sample. Then, the surface layer will be 2D raster-scanned by the 2-axis motorized stages, and all obtained tiles will be automatically stitched in parallel by our lab-developed programs based on MATLAB (MathWorks Inc); Step (3) during the sectioning mode, the motor-*y* will move leftwards to its original position, leaving enough space for vibratome sectioning. At the same time, the sample holder will move upwards at the same distance as the slicing thickness to maintain the sample surface always at the focal plane of the objective lens. The above three steps will be repeated until the whole organ has been completely imaged. To realize fully automated serial imaging and sectioning, a custom driving system based on a low-cost microcontroller unit (MCU) (Mega 2560, Arduino) has been developed to trigger and synchronize all hardware, including the 2-axis motorized stages, vibratome, and color camera. A corresponding control interface based on LabVIEW (National Instruments Corp.) has also been developed to adjust key scanning parameters such as the step size and travel range of the motorized stages.

## Sample preparation

Once mice (wild-type C57BL/6, 2 months old) were sacrificed, organs or embryos inside were harvested immediately and rinsed with phosphate-buffered saline (PBS) solution for a minute. Then, the organs or embryos will be submerged under 10% neutral-buffered formalin (NBF) at room temperature for 24 hr for fixation.

To achieve high sectioning quality, it is common to embed the tissue samples into agarose. For different types of tissues, the suggested concentration of the agarose and the corresponding working parameters of the vibratome will be varied (*Precisionary Instruments LLC, 2022*). To realize stable performance when sectioning hard organs (e.g., a fixed mouse spleen), 10% (w/v) gelatin is preferred for embedding because post-fixation by NBF overnight can tighten the connection between the sample surface and surrounding gelatin after crosslinking *Treiber et al., 1986*; *Griffioen et al., 1992*. Otherwise, the sample can be pulled out from the embedding medium during sectioning. Moreover, because of the porous structure of lung tissue, perfusing it with low-temperature agarose or gelatin through its trachea is necessary to provide adequate support inside the tissue.

All animal experiments were conducted in conformity with a laboratory animal protocol approved by the Health, Safety and Environment Office of the Hong Kong University of Science and Technology (HKUST) (license number: AEP16212921).

## Real-time molecular staining

Typically, a specimen should be washed after staining to remove residual unbonded probes on its surface and avoid strong fluorescence background. In TRUST, samples need to be immersed into the staining solutions during the whole experiment for real-time staining. Therefore, the dyes we used should have high specificity, and the intensity of the excited fluorescence signal should only be strong when the probes have been attached to the targets. Fluorogenic probes *Gao et al., 2020*; *Wang et al., 2020*; *Werther et al., 2021* show increased fluorescence upon target binding. The fluorogenic effect can substantially improve the signal-to-background ratio and make them widely applied for wash-free cell imaging experiments *Wieczorek et al., 2017*; *Grimm et al., 2015*; *Lukinavičius et al., 2013*. Obviously, fluorogenic probes are naturally suited for real-time staining. Lots of different fluorogenic probes have been synthesized for targeting various components, like proteins *Gao et al., 2020*; *Werther et al., 2021*; *Grimm et al., 2017*, lipid droplets *Fam et al., 2018*, cytoskeleton *Lukinavičius et al., 2014*, and mitochondrial *Fang et al., 2020*. DAPI and PI *Barcellona*

*et al., 1990*; *Unal Cevik and Dalkara, 2003* are two of the most commonly used and commercially available fluorogenic probes used together in TRUST for double-labeling to improve image color contrast and reveal more biological information (*Figure 1—figure supplement 1*). Since the staining speed of PI is much faster when compared with DAPI *Gong et al., 2016*, the staining time needed for each round can be simply quantified by analyzing changes in the total number of cells labeled by DAPI over time, which is approximately 150 s (*Figure 1—figure supplement 5*). Nile red and DiI are two common lipids staining dyes that are almost nonfluorescent in polar solvents (e.g., water) but undergo fluorescence enhancement in nonpolar environments. Their compatibility has also been tested with TRUST by imaging the brain and liver tissue without washing (*Figure 1—figure supplement 2*).

For small samples, the concentration of the staining solutions in the tank will not change significantly throughout the entire imaging process. However, for a large sample, e.g., a whole mouse brain, additional chemical dyes should be added into the water tank by a pump during imaging to maintain the concentration of the solution to be stable. The starting concentration of the staining solutions and the pumping rate applied for different samples are listed in *Supplementary file 1a*. Because the change of the concentration of solutions is relatively slow, in practice, the control of the pumping rate can be manually tuned based on the average brightness of the imaged sections. In the future, a negative feedback control system can be developed to automatically adjust the pumping rate according to the brightness of the imaged section in real-time. It is also possible to simply replace the water tank of the vibratome with one that can hold a larger volume of staining solutions.

## Image processing and statistical analysis

The electrical signal from MCU synchronizes the motor scanning and triggers the color camera to capture and save the current FOV as a binary file through its built-in program. A lab-built program based on MATLAB has been developed to further convert binary files to TIFF images and downsize them into different scales (4×, 25×, and 50×) in real-time to facilitate the low-resolution 3D rendering later. Also, the program will automatically start to stitch all mosaic FOVs since the current section of the sample has been fully scanned in order to monitor imaging results during the experiment. This feature could highly improve the image processing efficiency, considering the large size of the whole dataset (e.g.,~2 TB for a whole mouse brain).

Image registration is not necessary for stitching because relative positions between FOVs are well determined by the driving parameters of the high-precision stages. Another reason is that the cross-sections are mostly flat due to the reliable performance of the vibratome. To avoid a grid-like shading pattern on the stitched section due to uneven illumination, conventional flat-field correction *Seibert et al., 1998* and linear blending algorithm *Solina and Leonardis, 1999* have been applied. If necessary, notch filters in the frequency domain can be used to further eliminate the periodic stripes. The color dynamic range of images can also be further adjusted to enhance the contrast. The removal of the background surrounding the sample can be realized by manual masking to achieve high accuracy and minimize artifacts. The extraction of all glomeruli (*Figure 3—video 3*) in the 3D dataset of the mouse kidney was enabled with the assistance of a machine learning library, Detectron2 *Wu et al., 2019*, where TRUST images of two sections of a mouse kidney were manually labeled for the network training.

To render the 3D model, acquired 2D sections can be directly input to Amira (Thermo Fisher Scientific Inc) or ImageJ (NIH) *Schneider et al., 2012* with a z-axis interval set as the sectioning thickness. Limited by the computational power, a low-resolution mode is generally applied when rendering the whole block, and only a small zoomed-in region will be rendered in high resolution to show more details. 3D structures of some biological components can be extracted based on their colors or signal intensities. For example, nerve fibers in the brain appear to be bright in blue/green channels (*Figure 2t–v*), cell nuclei appear to be bright in the red channel (*Figure 2—figure supplement 1q–s*), and white pulp regions in the spleen appear to be bright in blue/green channel (*Figure 3v*). Also, the excited signal from vessels is obviously lower than that from surrounding tissues. Therefore, the negative contrast of the vessel network in organs (*Figure 2w, x*; *Figure 3j, o*) can be utilized for segmentation. All extracted structures have been rendered with pseudo color to enhance their visibility. Lookup tables of pseudo colors mostly represent the normalized intensity, except for *Figure 2w*, where the color varies with the axis perpendicular to the coronal plane.

## Code availability

The code for detectron2 is available at https://github.com/facebookresearch/detectron2.

## Acknowledgements

The Translational and Advanced Bioimaging Laboratory (TAB-Lab) at HKUST acknowledges the support of the Research Grants Council of the Hong Kong Special Administrative Region (16208620 and 26203619).

## Additional information

### Competing interests

Wentao Yu, Lei Kang, Yan Zhang: has applied for a patent (US Provisional Patent Application No.: 63/254,546) related to the work reported in this manuscript. Victor TC Tsang: has a financial interest in PhoMedics Limited, which, however, did not support this work. Ivy HM Wong: has a financial interest in V Path Limited, which, however, did not support this work. Terence TW Wong: has a financial interest in PhoMedics Limited, which, however, did not support this work. Has applied for a patent (US Provisional Patent Application No.: 63/254,546) related to the work reported in this manuscript.

### Funding

| Funder | Grant reference number | Author |
| --- | --- | --- |
| Research Grants Council, University Grants Committee | 16208620 | Terence TW Wong |
| Research Grants Council, University Grants Committee | 26203619 | Terence TW Wong |

The funders had no role in study design, data collection and interpretation, or the decision to submit the work for publication.

### Author contributions

Wentao Yu, Conceptualization, Data curation, Formal analysis, Validation, Investigation, Visualization, Methodology, Writing – original draft; Lei Kang, Software, Investigation, Visualization, Methodology; Victor TC Tsang, Resources, Investigation; Yan Zhang, Software, Visualization, Methodology; Ivy HM Wong, Resources, Visualization; Terence TW Wong, Conceptualization, Resources, Supervision, Funding acquisition, Methodology, Project administration, Writing – review and editing

### Author ORCIDs

Wentao Yu ⬤ http://orcid.org/0000-0002-4712-3177
Terence TW Wong ⬤ http://orcid.org/0000-0001-6399-758X

### Ethics

All animal experiments were conducted in conformity with a laboratory animal protocol approved by the Health, Safety and Environment Office of the Hong Kong University of Science and Technology (HKUST) (license number: AEP16212921).

### Decision letter and Author response

Decision letter https://doi.org/10.7554/eLife.81015.sa1
Author response https://doi.org/10.7554/eLife.81015.sa2

## Additional files

### Supplementary files

• MDAR checklist
• Supplementary file 1. Staining and camera parameters in TRUST, and the comparison of TRUST

with other whole-organ imaging modalities. (a) Parameters of the staining solutions and the white balance. Concentrations of the staining solutions and the rates for adding additional dyes into the water tank to maintain the same concentration of the staining solutions. Note: the control of pumping rate is not necessarily needed and can be different depending on the size of the specimen and volume of the staining solutions in the water tank. The white balance (red/blue) parameters of the color camera (DS-Fi3, Nikon Corp.) in TRUST are also provided as a reference. (b) Comparison of TRUST with other whole-organ imaging modalities. The imaging speed of each modality is calculated based on the reported literature for whole mouse brain imaging, including serial two-photon tomography (STPT) *Ragan et al., 2012*, microtomy-assisted photoacoustic microscopy (mPAM) *Wong et al., 2012*, micro-optical sectioning tomography (MOST) *Li et al., 2010*, fluorescence micro-optical sectioning tomography (fMOST) *Gong et al., 2013*, light-sheet fluorescence microscopy (LSFM) *Matsumoto et al., 2019*, and wide-field large-volume tomography (WVT) *Gong et al., 2016*.

## Data availability

Data availability. The authors declare that all data supporting the findings of this study are available within the paper and its supplementary information.

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
