## [Editor Report]

TRUST is a powerful content-rich three-dimensional imaging method. By combining iterative mechanical sectioning, automated labeling with fluorogenic dyes, UV-based surface illumination, and widefield multicolor detection, entire late gestation mouse embryos and terminally developed tissues can be imaged at cellular-scale resolution. Importantly, the strengths and weaknesses of the method are forthrightly presented. Ultimately, this work will be of great interest to developmental biologists, medical biologists, and clinicians, all of whom routinely work with specimens that are prohibitively large for labeling and imaging using classical mechanisms.

---

## [Decision Letter]

**Decision letter after peer review:**

Thank you for submitting your article "Translational Rapid Ultraviolet-excited Sectioning Tomography for Whole-organ Multicolor Imaging with Real-time Molecular Staining" for consideration by *eLife*. Your article has been reviewed by 2 peer reviewers, and the evaluation has been overseen by a Reviewing Editor and Didier Stainier as the Senior Editor. The reviewers have opted to remain anonymous.

As you will see below, both reviewers agreed that the presented technique is interesting and provides significant advantages over existing methods. As such, the reviewers found the paper appropriate for *eLife*. That being said, both reviewers thought that additional quantitative analyses were necessary to evaluate the merits and potential of the method. This includes the lateral and axial resolution, the penetration depth of the UV illumination, registration accuracy, the accuracy of the segmented results. Furthermore, greater effort could be made to discuss in the manuscript how this method is a significant advance over previous methods that also used ultraviolet surface excitation.

*Reviewer #1 (Recommendations for the authors):*

Specific Comments

1. Using a UV-LED as a light source can cause some problems. Although the authors claim that the speed is one advantage of TRUST, they still have a running time of tens of hours (few days). Doesn't the stability of the light source affect the uniformity of the entire image? Thus, UV can be hazardous to the sample being exposed. Are there any solutions to these anxiety factors? Following the description in Discussion, using multiple LEDs might exacerbate the problems.

2. In system configuration, the light was illuminated oblique. Why not configure it coaxially?

3. In organ-level imaging, important internal components have been visually extracted with a post-processing. However, in spleen imaging, vascular networks including veins and arteries were not reconstructed.

4. In this paper, the results were visualized and evaluated qualitatively. Authors are encouraged to provide the quantitative results. Especially for the case of whole mouse embryos, the biological changes over time can be quantified with various parameters.

5. Not only the soft tissues, can it be applied to hard tissues such as bones or teeth? Is there any difference in the solution supply and the vibratomy operation?

6. Is there any spatial error during the mechanical sectioning? If so, does it depend on the size and hardness of the sample or thickness of the slide? How do you correct this?

---

## [Author Response]

Reviewer #1 (Recommendations for the authors):Specific Comments1. Using a UV-LED as a light source can cause some problems. Although the authors claim that the speed is one advantage of TRUST, they still have a running time of tens of hours (few days). Doesn't the stability of the light source affect the uniformity of the entire image? Thus, UV can be hazardous to the sample being exposed. Are there any solutions to these anxiety factors? Following the description in Discussion, using multiple LEDs might exacerbate the problems.

Thanks for your valuable questions. On the one side, we believe that the potential of TRUST has not yet been fully exploited as mentioned in the *Discussion*, and the imaging speed of TRUST can be further improved by a factor of tens, thus reducing the UV illumination time. On the other side, regarding the stability of the light source in TRUST, the typical lifetime of the UV-LED (M285L5, Thorlabs Inc) used can be over 10,000 hours, which is much longer than the typical imaging time (tens of hours). From our experimental results, we also successfully acquired high-quality TRUST images with high uniformity, further validating the high stability of the UV-LED. Furthermore, we used the compatible and matching LED driver (LEDD1B, Thorlabs Inc) for stable and reliable control of the driving current of the LED.

Deep-UV light sources have been used in many biological imaging systems^1–3^. For example, TRUST is developed from MUSE^3–5^ which is also a method based on ultraviolet surface excitation. From the previous findings, the biological samples do not have any biochemical differences with or without MUSE staining and imaging^3^. The MUSE system with a dual-UV-LED setup has also been demonstrated previously^4^. Besides, although the imaging time for the whole mouse brain by TRUST can take tens of hours, due to the short penetration depth of the UV light, the exposure time of UV light for each imaged field of view (FOV) is short (~250 ms).

However, we agree that UV can still be hazardous to the sample being exposed, especially with multiple UV-LEDs*.* To relieve the issue while maintaining the high scanning speed, motors with high speed and/or cameras with a large chip size can be used. For example, by using the color camera (Digital Sight 10, Nikon) with a much larger chip size (35.8 mm ´ 23.8 mm) and with the help of super-resolution networks as mentioned in the *Discussion*, the total improvement factor can be over 150 times even with the original motors and one UV-LED setup.

2. In system configuration, the light was illuminated oblique. Why not configure it coaxially?

Thanks for your question. Coaxial illumination is commonly utilized in fluorescence microscopy, which facilitates the use of high NA and short working distance objective lenses in order to improve the achievable imaging resolution.

However, in TRUST, deep-UV light (~285 nm) is used as the excitation source and the excitation signals mostly fall in visible spectra (considering both the sensing spectra of the used color camera and also the applied fluorescent probes). Therefore, optical elements that perform well in both deep-UV and visible range are needed for coaxial configuration. However, these elements can be more expensive. Also, the working distance of the objective lens used in TRUST for subcellular imaging is fairly enough for the oblique illumination. If high NA and short working distance objective lenses are needed for better imaging resolution, sapphire light guides could be used to save the space needed for oblique illumination^6^. Moreover, as demonstrated previously^6^, the water immersion and the oblique illumination are helpful for further reducing the penetration depth of light (better axial resolution). Finally, due to the oblique illumination, traditional brightfield microscopes can be easily adapted to the 2D version of the TRUST system by simply adding a low-cost UV-LED, showing the high translational value of TRUST.

3. In organ-level imaging, important internal components have been visually extracted with a post-processing. However, in spleen imaging, vascular networks including veins and arteries were not reconstructed.

Thanks for your comment. In general, although much effort has been spent to reveal meaningful biological information in autofluorescence imaging^7–9^, relatively low imaging specificity is still one major limitation when compared with fluorescence imaging. For example, autofluorescence spectra of tissue components can overlap with each other and the signal intensity of individual components can be different even under the same excitation source. And that is the key reason why fluorescent dyes are also applied in TRUST for high-content imaging.

In the case of vascular networks, they can be well extracted when the surrounding tissue shows higher autofluorescence signal intensity than that of blood vessels^10^, as shown in Figure 3—figure supplement 1. However, in the case of spleen imaging, unfortunately, the background cannot provide sufficient contrast for the blood vessels’ digital extraction as shown in Figure 3x,y. Therefore, when high imaging specificity for vessels is required, instead of black ink perfusion as we demonstrated for brain vessel painting (Figure 2—figure supplement 3), the perfusion of fluorescent dyes as previously demonstrated^11^ should be used to realize better imaging specificity and ease the procedure for vessel extraction. Importantly, the perfusion protocol is compatible with TRUST and will not add much time cost.

4. In this paper, the results were visualized and evaluated qualitatively. Authors are encouraged to provide the quantitative results. Especially for the case of whole mouse embryos, the biological changes over time can be quantified with various parameters.

Thanks for your valuable recommendation. We agree that more effort should be made on quantitative analysis to fully explore the potential of TRUST. To show the potential of TRUST for facilitating biological study, for example, as shown in the newly added Figure 4—figure supplement 1, we can achieve volume quantification and comparison between embryos with different developmental stages. For simplicity of the demonstration, only two datasets were used. However, it already shows that TRUST is able to provide the possibility for developmental biology study.

5. Not only the soft tissues, can it be applied to hard tissues such as bones or teeth? Is there any difference in the solution supply and the vibratomy operation?

Thanks for your questions. Relieving researchers from laborious and time-consuming tissue preparation work is one of the goals that we are aiming for. We select the vibratome on purpose because the major of biological tissues are relatively soft and can be directly sectioned by vibratome. All vital organs in mice and even the skull/spinal cord in mouse embryos have been sectioned and imaged successfully using TRUST with high quality. In case, when extremely hard biological samples (e.g., bones in adult mice) are needed for sectioning, protocols (e.g., decalcification) can be applied to soften the specimen. Besides, with the overall working principle of TRUST unchanged, a microtome can be used to replace the vibratome for hard tissue sectioning.

We provided a table (supplementary file 1a) about the parameters of the staining solutions, including the pumping rate for two chemical dyes based on the organs. Replacing the water tank of the vibratome with one that can hold a larger volume of staining solutions can also keep the chemical concentration relatively constant. For different types of tissues, the operation parameters of the vibratome will be varied to achieve the best sectioning performance as suggested by *Precisionary Instruments Inc* (https://github.com/HEY-SAI/Cutting-parameters/files/9595073/Cutting.parameters.pdf).

6. Is there any spatial error during the mechanical sectioning? If so, does it depend on the size and hardness of the sample or thickness of the slide? How do you correct this?

Thanks for your questions. Compared with the microtome or cryostat, the advantage of the vibratome is that it can be used in serial sectioning-based 3D imaging systems^12–15^ for fixed or even fresh tissue, but we may need to pay more attention to achieve good slice quality.

For optimal sectioning quality, we directly integrated the high-end commercial vibratome (VF-700-0Z, Precisionary Instruments Inc) into TRUST. The slicing thickness and the hardness of the sample could certainly affect the sectioning quality. Therefore, although the claimed minimal sectioning thickness of the vibratome used is 10 µm, the sectioning thickness is mostly set as 50 µm to avoid imperfect sectioning and to produce even sections, which can also be shown in rendered supplementary videos (e.g., Figure 3-Video 1) where all sections of the heart have been rendered in high quality. In short, we did not observe any spatial error during the mechanical sectioning.

In addition, for samples with high toughness (e.g., fixed mouse heart or spleen), instead of agarose, 10% (w/v) gelatin is preferred as the embedding material because post-fixation by formalin overnight can tighten the connection between the sample surface and surrounding gelatin after cross-linking. Otherwise, the sample can be pulled out from the embedding medium during sectioning. As for porous tissue (e.g., lung tissue), perfusion or infiltration with low-temperature agarose or gelatin can provide adequate support inside the tissue after cooling and guarantee nice sectioning quality.

As for errors from spatial registration, lateral or axial registration of different layers is not necessary due to the high bidirectional repeatability (±0.2 µm) of the employed translation stages.

References

1. Zhang, Y. *et al.* High-Throughput, Label-Free and Slide-Free Histological Imaging by Computational Microscopy and Unsupervised Learning. *Adv. Sci.* 9, (2022).

2. Wong, T. T. W. *et al.* Label-free automated three-dimensional imaging of whole organs by microtomy-assisted photoacoustic microscopy. *Nat. Commun.* 8, 1386 (2017).

3. Fereidouni, F. *et al.* Microscopy with ultraviolet surface excitation for rapid slide-free histology. *Nat. Biomed. Eng.* 1, 957–966 (2017).

4. Xie, W. *et al.* Microscopy with ultraviolet surface excitation for wide-area pathology of breast surgical margins. *J. Biomed. Opt.* 24, 1 (2019).

5. Guo, J., Artur, C., Eriksen, J. L. and Mayerich, D. Three-Dimensional Microscopy by Milling with Ultraviolet Excitation. *Sci. Rep.* 9, (2019).

6. Yoshitake, T. *et al.* Rapid histopathological imaging of skin and breast cancer surgical specimens using immersion microscopy with ultraviolet surface excitation. *Sci. Rep.* 8, (2018).

7. Ojaghi, A. *et al.* Label-free hematology analysis using deep-ultraviolet microscopy. *Proc. Natl. Acad. Sci. U. S. A.* 117, 14779–14789 (2020).

8. Bhartia, R. *et al.* Label-free bacterial imaging with deep-UV-laser-induced native fluorescence. *Appl. Environ. Microbiol.* 76, 7231–7237 (2010).

9. Kaza, N., Ojaghi, A., Casteleiro Costa, P. and Robles, F. E. Deep learning based virtual staining of label-free ultraviolet (UV) microscopy images for hematological analysis. in Label-free Biomedical Imaging and Sensing (LBIS) 2021 (eds. Shaked, N. T. and Hayden, O.) vol. 11655 13 (SPIE, 2021).

10. Mehrvar, S. *et al.* Three-dimensional vascular and metabolic imaging using inverted autofluorescence. *J. Biomed. Opt.* 26, 1–16 (2021).

11. Henriksen, B. L. E., Jensen, K. H. R. and Berg, R. W. Vasculature-Staining with Lipophilic Dyes in Tissue-Cleared Brains Assessed by Deep Learning. *SSRN Electron. J.* (2020) doi:10.2139/ssrn.3606784.

12. Economo, M. N. *et al.* A platform for brain-wide imaging and reconstruction of individual neurons. *eLife* 5, (2016).

13. Zhou, C. *et al.* Continuous imaging of large-volume tissues with a machinable optical clearing method at subcellular resolution. *Biomed. Opt. Express* 11, 7132 (2020).

14. Min, E. *et al.* Serial optical coherence microscopy for label-free volumetric histopathology. *Sci. Rep.* 10, 6711 (2020).

15. Abdeladim, L. *et al.* Multicolor multiscale brain imaging with chromatic multiphoton serial microscopy. *Nat. Commun.* 10, 1–14 (2019).

16. Niu, J. *et al.* Propidium iodide (PI) stains Nissl bodies and may serve as a quick marker for total neuronal cell count. *Acta Histochem.* 117, 182–187 (2015).

17. US11346782B2 – Tomographic imaging method – Google Patents. https://patents.google.com/patent/US11346782B2/en?inventor=Shaoqun+ZENGandoq=Shaoqun+ZENG.

18. Wong, T. T. W. *et al.* Label-free automated three-dimensional imaging of whole organs by microtomy-assisted photoacoustic microscopy. *Nat. Commun.* 8, 1386 (2017).